# Competition-Induced Binding Spherical Nucleic AcidFluorescence Amplifier for the Detection of Di (2-ethylhexyl) Phthalate in the Aquatic Environment

**DOI:** 10.3390/nano12132196

**Published:** 2022-06-26

**Authors:** Lin Yuan, Dandan Ji, Qiang Fu, Mingyang Hu

**Affiliations:** College of Environmental Science and Engineering, Qilu University of Technology (Shandong Academy of Science), Jinan 250353, China; m17862975991@163.com (L.Y.); micro7306@163.com (Q.F.); jnhmy17854118089@163.com (M.H.)

**Keywords:** competition-induced, aptamer-walker, spherical nucleic acid, Di-2-ethylhexyl phthalate, fluorescence

## Abstract

Di-2-ethylhexyl phthalate (DEHP) is a toxic plasticizer and androgen antagonist. Its accumulation in water exceeds national drinking water standards and it must be continuously and effectively regulated. Currently, methods used to detect DEHP are still unsatisfactory because they usually have limited detection sensitivity and require complex operating procedures. A competition-induced fluorescence detection method was developed for the selective detection of DEHP in an aquatic environment. An aptamer with walking function was used as the recognition element for DEHP, and its quantification was induced by competition to change the fluorescence signal. The detection range was 0.01~100 µg/L, and the detection limit was 1.008 μg/L. This high-sensitivity DEHP detection capability and simplified process facilitates real-time fields and other monitoring tasks.

## 1. Introduction

Di-2-ethylhexyl phthalate (DEHP) is a toxic plasticizer made from octanol or isooctanol and phthalic acid [1]. DEHP has good compatibility with most industrial synthetic resins and rubbers. Because it can be widely used as food packaging and medical packaging additive, it may accumulate in water and exceed the national standards for drinking water [2,3]. At the same time, it is also a type of androgen antagonist. Long-term accumulation in the human body may cause endocrine disorders and immunity decline, and may produce carcinogenic reactions in animals [4]. Therefore, it is necessary to perform continuous and effective monitoring and management of DEHP.

The detection methods of DEHP mainly include chromatographic analysis, e.g., gas chromatography (GC) [5] and high-performance liquid chromatography (HPLC) [6,7], immunoanalysis and biosensor analysis [8,9]. Currently, the most widely used chromatographic analysis methods (GC and HPLC) are still dominant in the current detection field because of their high sensitivity, accuracy and reproducibility [10]. However, because such detection methods require expensive large-scale instruments and equipment, professional and technical personnel to operate accurately, with the disadvantages of high analytical costs and long analysis cycles, the use of instruments and equipment is limited, and cannot achieve rapid on-site detection of one or more pollutants. At the same time, due to the water environment samples, many interfering impurities, high instrumentation injection standards, and the need for pre-processing (de-hybridization, purification, concentration) of the collected water samples, greatly increase the difficulty of detection. At the same time, the immunoassay method has some problems, such as long preparation time of antibody, false-positive recognition of target molecule by antibody, and difficulty with in vitro preservation. Therefore, it is very important to develop a fast, portable, and stable analytical method. Aptamers are short sequence oligonucleotides or polypeptides obtained by in vitro screening [11,12]. Aptamers have properties such as easy modification, high affinity and resistance against denaturation, which provide a new research method for efficient and rapid recognition of small molecular targets [13].

Meanwhile, in a stable system where the aptamer coexists with the target and complementary DNA, the molecular interaction forces resulting from its interaction with the target cause it to bind preferentially to the target to form the aptamer-target complex [14]. For instance, Lim et al. demonstrated that in the presence of DEHP, the aptamer and the DNA probe form a double-stranded structure which is specifically induced to dissociate by DEHP to form a more robust DEHP-aptamer structure [12]. Considering that the concentration of DEHP in natural water samples is trace, a stable signal amplification pathway needs to be developed to improve the sensitivity of the assay. Spherical nucleic acid (SNA) is a type of nanometer element composed of a highly oriented and highly dense oligonucleotide layer, which can be used as a stable signal amplification element [15,16,17]. Mason et al. designed a biomolecule−nanoparticle interface using stochastic DNA walkers and demonstrated the critical roles of varying interfacial factors, including intraparticle interaction, orientation, cooperativity, steric effects, multivalence, and binding hindrance [15]. The persistent movement of DNA walkers and subsequent enzymatic reactions are mediated by the exquisite Watson-Crick base-pairing and, thus, are highly predictable and programmable [15,18]. In addition, Lu et al. isolated an aptamer specific for DEHP from immobilized ssDNA library, and the aptamer has the highest affinity (Kd = 2.26 ± 0.06 nM) [19]. Using the high specificity of the aptamer, we designed a competition-induced fluorescence amplifier detection method to quantify DEHP in the aquatic environment, in which the aptamer was added to the walking function to act as both a recognition element for the target contaminant and an amplifying element for the competition-induced fluorescence signal. SNA was used as a functional carrier for the continuous and stable release of the fluorescence signal.

## 2. Experimental Section

### 2.1. Chemicals

DEHP, dibutyl phthalate (DBP), dioctyl phthalate (DOP), diisodecyl phthalate (DPHP), and butyl benzyl phthalate (BBP) were purchased from Sigma-Aldrich (Saint Louis, MO, USA). Naphthalene, triphenylmethane, HgCl_2_, CdCl_2_, CoCl_2_ and H_3_AsO_3_ were purchased from Sigma-Aldrich (Saint Louis, MO, USA). Nt. BbvCI and 10 × CutSmart buffer were purchased from New England Biolabs (Beijing, China). The DNA sequences were ordered from Takara Biotechnology Co., Ltd. (Dalian, China) with HPLC purification, and they are shown in Table 1.

### 2.2. Apparatuses

All fluorescence assay measurements were performed on Tecan Infinite 200pro (Tecan, Switzerland). UV absorption spectra were measured using a UV spectrophotometer (Shimadzu UV-1750). Transmission electron microscopy (TEM) images of AuNPs were obtained using a JEM-3010 transmission electron microscope. HPLC data were measured using LC20 (Japan, Shimadzu).

### 2.3. Preparation of SNA

AuNPs (13 nM) were first prepared by the sodium citrate reduction method [20], and their TEM image is shown in Appendix A. The maximizing loading of the DNA oligonucleotides method established by Li et al. was slightly modified to prepare SNA [21]. First, 50 μM tracks were mixed with 1 nM of 1 mL AuNPs. This mixture was left at room temperature for 10 h and then slowly mixed with 20 μL of 3 M NaCl, followed by 10 s of sonication. This process of adding NaCl and sonication was repeated 5 times with 1 h intervals to maximize the loading of tracks. The solution was left to stand at room temperature for 24 h. Then, the solution was centrifuged at 12,000 rpm for 30 min to separate the track-functionalized AuNPs. SNA was washed 3 times with a 1× TE buffer (PH = 8) and finally redispersed in 1× TE buffer (PH = 8). The 1× TE buffer concentration composition had 10 mM Tris-HCl and 1 mM EDTA. The TEM image is shown in Appendix A. The UV absorption spectra of AuNPs and SNA are shown in Appendix A.

### 2.4. Fluorescence Assay

All DNA strands were heated at 95 °C for 5 min, placed at 0 °C for 5 min, and incubated at room temperature for 1 h before use. In a typical reaction, a mixture containing 2.5 nM aptamer-walker, varying concentrations of the target molecule (DEHP, DBP, DOP, DPHP, BBP and other distractors), and 1× TE buffer was incubated at 37 °C for 1 h. Then, 0.5 nM AuNP-tracks, 0.2 U/μL Nt. BbvCI, and 1× Cut Smart buffer were added to the mixture. The 1× CutSmart buffer concentration composition contained 50 mM potassium acetate, 20 mM Tris-acetate, 10 mM magnesium acetate, and 100 µg/mL bovine serum albumin (BSA). The reaction mixture was added to a 96-well microplate. Fluorescence was measured directly using a multimode microplate reader with excitation at 535 nm and emission at 595 nm. The fluorescence increase was measured every 0.5 min for the first 10 min and then every 1 min for another 20 min. Additionally, a reaction system without DEHP was established as a negative control to determine the feasibility of the reaction.

### 2.5. Optimal Assay

In the reaction system, the doses of aptamer-walker or Nt. BbvCI, the combination time and temperature with an aptamer-walker and DEHP, as well as the optimal reaction temperature and time, will affect the sensitivity of the reaction. The optimal factors were determined by univariate experiments. Among them, the optimal concentration range of aptamer-walker was 0~10 nM. The optimal concentration range of Nt. BbvCI was 0~0.5 U/μL; the combination time of the aptamer-walker with DEHP was 15~150 min, and the binding temperature of the aptamer-walker with DEHP was 31~43 °C.

### 2.6. Detection of PCB 77 in Water Samples

In the actual sample study, a water sample of mixed wastewater was used as the research object. The wastewater sample contained DEHP, DBP, DOP, naphthalene, and heavy metal ions (Hg^2+^, Cd^2+^ and Co^2+^), which were detected in selectivity experiments. Among them, the amount of DEHP standard product was 50 µg/L. First, the mixed water samples were filtered to remove suspended pollutants and eliminate their interference with fluorescence signals. Then, the filtered mixed water samples (10 µL) and different concentrations of DEHP were simultaneously added with an aptamer-walker in TE buffer. The final concentration range of the DEHP standard in TE buffer was 0, 1, 2, 4, 8, 12, 16, and 20 µg/L. According to the detection steps in the abovementioned fluorescence analysis method, the change value of the fluorescence signal was measured, and a calibration curve was established.

### 2.7. Detection of PCB 77 in Water Samples by HPLC

The manually configured wastewater samples need to be treated as follows. It is necessary to pretreat the wastewater sample. First, 100 mL of the wastewater sample was filtered through a 0.45 µm microporous organic filter membrane, and phosphoric acid or ammonia water was added to adjust the pH of the water sample to neutral. A total of 2 mL of methanol solution was added 3 times to activate the SPE column, and then methanol was eluted with deionized water. Then, the water sample was passed through the SPE column at a flow rate of 1 mL/min. Then, the samples were dried with nitrogen, and 1 mL of 5% methanol solution was slowly added. Finally, the SPE column was dried with purified air for 10 min. DEHP, which passed through the SPE column, was eluted 3 times with 10 mL of n-hexane solution, and the eluent was collected, concentrated to 1 mL, and quantified for subsequent experiments. The chromatographic conditions were as follows: chromatographic column (YWG-G18 column: 250 mm × 4.6 mm); mobile phase (95% acetonitrile); flow rate (1.0 mL/min).

## 3. Results and Discussion

### 3.1. Principle of the Competition-Induction Detection System

As shown in Figure 1, the aptamer-walker structure consists of two parts, a neck ring in the middle for DEHP recognition, whereas the 3′ and 5′ ends are connected to eight “foot” chains with the same sequence of bases for walking on SNA [16]. In the absence of DEHP, the aptamer-walker remained stable and closed. In the absence of DEHP, the aptamer-walker, SNA, and Nt. BbvCI were present simultaneously, and the “foot” strand is paired with the tracks (with fluorescent groups). Due to the enzymatic cleavage of Nt. BbvCI, the tracks caused the fluorescent groups to fall off SNA. Because fluorescent groups are spatially displaced from SNA, this results in a fluorescent signal change. Because the tracks break and the “foot” chains of its complementary pairing aptamer-walker fall off and bind to the next adjacent tracks, all tracks circulating to SNA are dropped. Under the condition of DEHP, the aptamer-walker and DEHP produce specific binding to form stable cluster polymers [15]. Because the polymer formed by the two is more stable compared to the binding of the “foot” to the track, it prevents the “foot” strand from walking on the surface of AuNPs. After the addition of Nt. BbvCI, SNA remains in its original stable state and produces no fluorescence signal change.

### 3.2. Feasibility and Characterization

The results of verifying the feasibility of the reaction are shown in Figure 1. The fluorescence spectra show high background values without DEHP (in black). Under the condition of 20 µg/L DEHP, an aptamer-walker is a cluster polymer formed by a double-stranded structure being opened and folded with DEHP. The cluster polymers formed by DEHP and aptamer-walkers are relatively stable in spatial structure, so that the “foot” domains at the ends of the aptamer-walker cannot walk freely. Nt. BbvCI has no specific cleavage site without cleavage reaction; tracks remain intact, and the fluorescent group cannot fall off from SNA, which results in the absence of change in the fluorescence signal (in red). Because the reaction process is a dynamic process, the fluorescence signal value will reach the maximum fluorescence value at approximately 10 min and will not continue to increase. The fluorescence intensity value of 15 min was used in the subsequent experiments to ensure the completion and evaluation of the reaction.

### 3.3. Optimization Assay

In this reaction system, the addition dosage of aptamer-walker, Nt. BbvCI, the combination time with aptamer-walker and DEHP, and the combination temperature with aptamer-walker and DEHP all affect the overall sensitivity of the reaction. These best factors of the reaction system can be determined by a single factor test. As shown in Figure 2A, in the presence of DEHP, with an increase in the concentration of aptamer-walker, the fluorescence signal value shows an increasing state. However, the concentration of SNA in the reaction system is certain with the content of DEHP. When the addition of aptamer-walker reaches 2.5 nm, the fluorescence signal value will no longer increase; thus, it can be determined that the optimal concentration is when the concentration of aptamer-walker is 2.5 nM. However, the Nt. BbvCI changes in enzyme concentration and fluorescence signal also increase with increasing enzyme concentration and maintain a stable state after reaching a certain limit. Therefore, it is determined that Nt. BbvCI is 0.2 U/μL (Figure 2B). Meanwhile, there are also important factors affecting the fluorescence signal change value of the reaction system (Figure 2C,D). Because the specific combination stability of aptamer-walker and DEHP is related to its combination time and combination temperature, there will be less free aptamer-walker in the reaction system with higher combination degree. The number of free aptamer-walkers will determine the change in the fluorescence signal value. Therefore, the more aptamer-walker there was in the TE buffer, the stronger the fluorescence signal value. Therefore, the optimal combination time of aptamer-walker and DEHP is 60 min, and the optimal combination temperature of aptamer-walker and DEHP is 37 °C.

### 3.4. Quantitative Analysis

To determine whether the reaction system can quantitatively detect DEHP in a water environment, the fluorescence change curves of DEHP at different concentrations are obtained. As shown in Figure 3, the fluorescence signal changes in the reaction system measured at different concentrations of DEHP are in the range of 0~100 µg/L. The change value of the fluorescence signal exhibits a linear change of two segments. When the DEHP concentration was in the range of 0~20 µg/L, the change in the fluorescence signal showed an exponential correlation with an increase in the DEHP concentration. This was converted into a linear curve y = 7.78 + 8.34x with a correlation coefficient of R^2^ = 0.998. The limit of detection (LOD) was 1.008 µg/L (3σ/slope rule). When the DEHP concentration was in the range of 20~100 µg/L, the change value of the fluorescence signal showed a linear curve correlation with an increase in the DEHP concentration, a linear curve y = 34.01 + 0.57x, and a correlation coefficient R^2^ = 0.998. Therefore, compared with previously reported DEHP detection methods, the proposed detection method has good advantages (in Table 2). The results of this work are better than those of most current detection methods.

### 3.5. Selectivity Analysis

DEHP usually does not exist alone in polluted aquatic environments, and there are often homologous interferences and other environmental interferences. To avoid the influence of other substances on the test results, specific experiments are carried out to test the accuracy of the experimental results. As shown in Figure 4, four structural analogues (DBP, DOP, DPHP, and BBP) with equal doses were first selected to evaluate the selectivity of the reaction system. Compared with the blank group, the DEHP fluorescence signal had the highest fluorescence change value, and the DBP, DOP, DPHP, and BBP fluorescence signal change value was 3/4 that of DEHP. In the reaction system for the determination of structural analogues, these four analogues all produce certain fluorescence signal changes, which may be due to their similarity to DEHP chemical structure. However, the chemical structure and molecular weight of naphthalene and triphenylmethane differ greatly from DEHP, and they produce little change in the fluorescence signal value. Meanwhile, to avoid the interference of heavy metal ions or metal-like ions with the experimental selectivity, Hg^2+^, Cd^2+^, Co^2+^, and As^3+^ were measured, and these ions also produced little change in the fluorescence signal value. These results indicate that the binding and selectivity of PAEs by this aptamer-walker need to be further improved. To use the reaction as a biological detection system for pollutant analysis in the water environment, its specificity must be greatly improved.

### 3.6. Water Sample Analysis

To further evaluate the detection accuracy of the system in contaminated water samples, the concentration of DEHP in manually configured wastewater samples was measured using the established detection method, and the detection accuracy was determined by adding standard and recycling methods. As shown in Figure 5, within the concentration range of 0~20 µg/L DEHP, with an increase in DEHP concentration, the relative fluorescence change value showed a good exponential correlation with DEHP concentration. The linear equation was y = 9.12x + 23.17, and the correlation coefficient was R^2^ = 0.998. According to the method of labelled recovery, the concentration of DEHP was calculated to be 5.82 µg/L because 10 µL of contaminated water was measured, and the total volume of fluorescence was 100 µL. Therefore, the concentration of DEHP in the artificially prepared wastewater sample was 58.2 µg/L, and the recovery rate was 116.4%. The method can be used to quantitatively determine the content of DEHP in actual wastewater samples.

### 3.7. Water Sample Analysis by HPLC

The different concentrations of DEHP standard (0, 5, 10, 20, 50, 100 g/L) were determined according to the abovementioned HPLC method, and a standard curve was drawn according to the relationship between the peak area and concentration of the chromatographic peaks of DEHP. As shown in Figure 6, the peak area was directly proportional to the concentration, and the linear relationship was y = 48.63 + 156.56x, R^2^ = 0.998. According to the abovementioned HPLC method, the concentration of DEHP in the same manually configured wastewater was determined. Because the concentration of DEHP in the actual sample after concentration was too high, a 10 µL concentrated sample was taken and diluted to 1 mL with n-hexane solution for determination. The peak area was 8191; the DHEP concentration was calculated to be 52.01 µg/L, and the recovery rate was 104.02%. Compared with this method, the accuracy of DEHP detection by HPLC was relatively higher, but its complex pretreatment process required considerable time and cost. This method only requires filtering the water sample before the analysis, which is cheap and convenient and is more suitable for rapid detection of DEHP in a water environment.

## 4. Conclusions

In conclusion, a fluorescence assay for the detection of DEHP by competition induction combined with SNA validated its performance. In this study, the aptamer of DEHP was modified by adding two “feet” of the same sequence, which increased the modifiability of the aptamer. The reaction process was induced by competitive interactions, and the concentration of the target contaminant was quantified by value changes in the fluorescence signal. At the same time, the stable fluorescence signal output by SNA made the system more suitable for the detection of trace amounts of DEHP in aquatic environments. In contrast to traditional chromatographic methods that require complex extraction and concentration of wastewater samples, this system only requires filtration of the wastewater samples, and the concentration of the sample to be measured can be determined by the fluorescence signal change response, saving processing cost and time. In subsequent experiments, the versatility, stability, and portability of the reaction system will be improved to make it more suitable for field testing of actual water environment samples.

## Data Availability

Not applicable.

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
