# Peer review of "Competition-Induced Binding Spherical Nucleic AcidFluorescence Amplifier for the Detection of Di (2-ethylhexyl) Phthalate in the Aquatic Environment"

_nanomaterials, 2022, doi:10.3390/nano12132196_

Round 1
Reviewer 1 Report
The work is interesting and well presented. It appears suitable for publication upon the following modifications:
- Rework the introduction in order to include references to the walking function on one hand and on the competetion-induced fluorescence employed in other studies with the same purpose.
- Please include in the experimental part the details of Cutsmart buffers.
- Experimental, L84 - please describe TE buffer.
- Please support section 3.1 with reference entries.
Reviewer 2 Report
I find the paper very nice and interesting. However, authors should better explain what are the methods that are currently used. They only say it is very expensive chromatorgraphic methods, but do not give further details. It should be better explained in order to understand the advantages of tje methond proposed in the paper.
